Integrative machine learning analysis of multiple gene expression profiles in cervical cancer

Tan Mei Sze 1
Chang Siow-Wee siowwee@um.edu.my 1
Cheah Phaik Leng 2
Yap Hwa Jen 3
1 Bioinformatics Programme, Institute of Biological Sciences, Faculty of Science, University of Malaya , Kuala Lumpur , Malaysia
2 Department of Pathology, Faculty of Medicine, University of Malaya , Kuala Lumpur , Malaysia
3 Department of Mechanical Engineering, Faculty of Engineering, University of Malaya , Kuala Lumpur , Malaysia
Jun Goo
Electronic publication date: 2018 Jul 25
Publication date: 2018
Volume: 6
Electronic Location ID: e5285
Received 2018 Mar 1; Accepted 2018 Jul 2
Copyright: ©2018 Tan et al.
Copyright year: 2018
Copyright holder: Tan et al.
License: This is an open access article distributed under the terms of the Creative Commons Attribution License, which permits unrestricted use, distribution, reproduction and adaptation in any medium and for any purpose provided that it is properly attributed. For attribution, the original author(s), title, publication source (PeerJ) and either DOI or URL of the article must be cited.
License URL: https://creativecommons.org/licenses/by/4.0/

Keywords: Gene expression profiling, Meta-analysis, Machine learning, Feature selection, Cervical cancer prognosis, Potential gene signature

Funding: University of Malaya RP038C-15AET BK041-2014 This study was supported by the University of Malaya research grants with the project number of RP038C-15AET & BK041-2014. The funders had no role in study design, data collection and analysis, decision to publish, or preparation of the manuscript.

==============================
Although most of the cervical cancer cases are reported to be closely related to the Human Papillomavirus (HPV) infection, there is a need to study genes that stand up differentially in the final actualization of cervical cancers following HPV infection. In this study, we proposed an integrative machine learning approach to analyse multiple gene expression profiles in cervical cancer in order to identify a set of genetic markers that are associated with and may eventually aid in the diagnosis or prognosis of cervical cancers. The proposed integrative analysis is composed of three steps: namely, (i) gene expression analysis of individual dataset; (ii) meta-analysis of multiple datasets; and (iii) feature selection and machine learning analysis. As a result, 21 gene expressions were identified through the integrative machine learning analysis which including seven supervised and one unsupervised methods. A functional analysis with GSEA (Gene Set Enrichment Analysis) was performed on the selected 21-gene expression set and showed significant enrichment in a nine-potential gene expression signature, namely PEG3, SPON1, BTD and RPLP2 (upregulated genes) and PRDX3, COPB2, LSM3, SLC5A3 and AS1B (downregulated genes).

Introduction

Cervical cancer, ranked 4th most common cancer (World Cancer Research Fund International, 2017) among women worldwide, caused 279,311 mortality cases in the year 2015 (World Health Organization, 2016). It is the 3rd most common cancer (7.7%) among Malaysian females (AbM, Nor Saleh & Noor Hashima, 2016). Most of the development of cervical cancer is linked to the infection of the Human Papillomavirus (HPV) that passes to the human host during sexual activity. The other mentioned risk factors include smoking, imbalanced diet and family history (Plummer et al., 2003; Ames & Gold, 1997; Lee et al., 2003). However, as in many cancers, it is conceivable that genetic pathways play a role in the development of cervical cancer.

Microarray technology has been used to simultaneously identify a large number of gene expressions that may be associated with a particular cancer. Reports of prognostically predictive genes have been identified by whole genome microarray in cancer. Hu et al. (2010) showed that miR-200a and miR-9 were associated with cervical cancer outcome while Aziz et al. (2016), reported on 19 common genes that were predictive in colorectal cancers. Meta-analysis is touted to solve the problems of low statistical power of studies, to be able to analyze intersected genes between datasets, extract the optimum values across multiple datasets and identify the most commonly expressed genes (Warnat, Eils & Brors, 2005; Brazma et al., 2001). Based on our literature review, we believed that this is one of the first few studies which involves meta-analysis using multiple cervical cancer microarray datasets.

Despite the many approaches which were introduced in the previous meta-analysis study, a biased issue that is caused by merging datasets from different platforms and experiments is a hot topic in the field. Alles et al. (2009), Chow, Alias & Jamal (2017), Dong et al. (2010) and Li et al. (2016) carried out meta-analysis in various type of cancers to find the significant genes in certain diseases such as breast cancer, paediatric B-acute lymphoblastic leukaemia (B-ALL), ovarian cancer, etc. Trendily, most of these studies were done by using the original datasets that varied from different platforms. As the original datasets contained noisy data such as the extreme values, thus the bias effects from each of the respective datasets might be strong.

Furthermore, until the present, most of the previous studies either used statistical analysis methods (Alles et al., 2009; Chow, Alias & Jamal, 2017; Dong et al., 2010; Li et al., 2016) or machine learning-based methods (Chow, Alias & Jamal, 2017; Dong et al., 2010; Li et al., 2016; Grützmann et al., 2005; Liu et al., 2004; Obayashi & Kinoshita, 2009; Wang & Makedon, 2004; Díaz-Uriarte & De Andres, 2006) in carrying out the meta-analysis. Due to different measurements being used in analysing the expression activities in the genes, there is no way in comparing the feasibility of both methods in a meta-analysis. Hence, by combining both the statistical and machine learning methods, meaningful insights could be discovered. Favourable features from both of the methods are integrated and fully utilized in carrying out the meta-analysis in this study.

To address the shortcomings of meta-analysis we attempted to build a system which integrates machine learning with meta-analysis to improve the quality of the data acquired. The whole process encompasses (i) gene expression analysis of individual dataset, (ii) meta-analysis of multiple datasets, and (iii) feature selection (FS) & machine learning analysis. We report our findings using our proposed integrative analysis in identifying genes which could be integral in the development of cervical cancers, a cancer in which understanding has moved significantly forward and awaits unravelling of possible genetic intermediaries in the carcinogenetic process.

Materials and Methods

In brief, our approach commences with the gene expression analysis of individual cervical cancer microarray datasets obtained from publicly available sources and to detect the genes that perform most significantly in each of the dataset. The most significant gene sets from each dataset are combined and intersected to form a larger dataset in the first stage. Microarray gene expression meta-analysis is performed using the combined dataset in the second stage. For the final stage, several FS and machine learning methods as well as a functional analysis by gene set enrichment analysis (GSEA) will be carried out to identify a more precise set of potential gene markers in cervical cancer biology.

Microarray datasets

All the data used are from the public website National Center Biotechnology Information Gene Expression Omnibus (NCBI GEO) (http://www.ncbi.nlm.nih.gov/geo). A search using the keywords “cervical cancer” and “microarray” was carried out. To ensure the accurateness and purity of the resultant 104 (before 5th January 2017) datasets, filtration was performed by manually reviewing each dataset using the following criteria. To gain entry into this study, the dataset must (1) contain expression profiling by array types, (2) use CEL raw files, and (3) be studies that involve cervical cancer as the main subject. Finally, only four cervical cancer microarray gene expression datasets (see Table 1) could be entered for further analysis.

Table 1 Details of the four microarray datasets used in this research.

Study	Platform	Samples	Genes	Variables	
Noordhuis et al. (2011) (GSE26511)	HG-U133_Plus_2	39	54,675	Lymph node status: negative, positive{19}	
Bachtiary et al. (2006) (GSE5787)	HG-U133_Plus_2	33	54,675	Biopsy: 1{5}, 2{6}, 3{7}, 4{5}, 5{5}, 6{3}, 7{2}	
Scotto et al. (2008) (GSE9750)	HG-U133A	61	22,283	tissue: normal{24}, cell line{5}, cancer{28}	
Den Boon et al. (2015) (GSE63514)	HG-U133_Plus_2	130	54,675	tissue: normal{24}, CIN1 lesions{14}, CIN2 lesions{22}, CIN3 lesions{40}, cancer{28}	

All analyses of the data were performed in R Core Team (2016).

Integrative approach of meta-analysis and machine learning

The aim of the proposed integrative approach is to fully utilize the benefits from both of the statistical analysis and the machine learning approaches in carrying out gene expression meta-analysis, in order to find out the most significant gene markers in cervical cancer. Generally, the proposed integrative analysis is composed of three stages, namely: (i) gene expression analysis of individual dataset; (ii) meta-analysis of multiple datasets and (iii) feature selection and machine learning analysis. The proposed integrative method is illustrated in Fig. 1. At the first stage, gene expression analysis of individual dataset was carried out where 10,000 differentially expressed genes from each dataset were selected by using Linear Models for Microarray Analysis (LIMMA). This process was repeated on four datasets. In stage (ii), meta-analysis of four datasets was carried out. Intersection process was done in order to obtain the intersected genes among the four datasets. A set of upregulated and downregulated genes were selected and ranked accordingly by using Rank Prod at the end of stage (ii). In stage (iii), eight FS and machine learning approaches were applied in order to identify the common optimum genes.

Figure 1 Framework for the proposed integrative approach of meta-analysis and machine learning in gene expression profiling of cervical cancer.

*DE- differential expression value

Stages of the integrative approach are as detailed below:

Stage (i): gene expression analysis of individual dataset

The data were first filtered to select out cancer samples (removal of normal samples) to ensure that the activities of the genes presented in the normal samples would not affect the analysis. Among the four sets of microarray data used in our study, only microarray data of GSE9750 (Scotto et al., 2008) is based on the Affymetrix Human Genome U133A Array [HG-U133A] platform. The other three datasets are based on Affymetrix Human Genome U133 Plus 2.0 Array [HG-U133_Plus_2]. Therefore, it is necessary to pre-process and normalize all of the raw level probe data across the platforms to reduce bias caused by the different platforms. A quantile algorithm was applied to normalize the data and the data were log2-transformed according to all the samples median.

Then, gene expression analysis was performed on each of the microarray datasets. RMA normalization from affy function in Bioconductor package in R (Gentleman et al., 2004) was carried out for every sample in the dataset to create expression values that are normalized and background corrected.

After that, differential gene expression analysis was carried out to select the top 10,000 differentially expressed genes from individual dataset using Linear Models for Microarray Analysis (LIMMA) (Smyth, 2004). In this stage, the rare and over-extremely expressed genes in every dataset were eliminated by using moderated t-statistics of LIMMA so that bias errors from each dataset could be reduced when combining the datasets to become a larger dataset.

The R package limma (Ritchie et al., 2015) was used to rank and select out the top differential expressed genes. The differentially-expressed genes were filtered using adjusted p-value of <0.05 and ranked based according to the p-values. In this study, the same gene expression analysis was repeated on all the four datasets.

Hence, this stage is treated as quality control of the dataset which is going to use in stage (ii) analysis as genes produced by this stage are consistently different between the group according to the LIMMA statistics.

Stage (ii): meta-analysis for multiple datasets

The aim of this stage is to select the genes that show upregulated and downregulated activities in cervical cancer. Four microarray gene expression datasets that contained top 10,000 genes each (selected in stage(i)) were combined in the intersection process by identifying the intersected genes found among the four datasets and merging all samples to form a single, larger sample size dataset (meta-dataset).

Next, meta-analysis was conducted by using RankProd (Breitling et al., 2004) package in R to analyse the intersected genes. With the default value of p < 0.05, FDR < 0.05 as the cut-off point, RankProd ranked the genes into two group: upregulated (FC < 1, php < 0.01) and downregulated genes (FC < 1, php < 0.01).

Stage (iii): feature selection (FS) and machine learning methods

Stage (iii) is to select the optimum genes to ensure the final set of genes selected is with higher standard and high quality in term of genetic evaluation. This is also to fully utilize the measurements of different machine learning methods (supervised and unsupervised) by collaborating them in selecting the significant genes. The machine learning methods chosen in this study are based on the literature review done which obtained good result in the related studies (Urbanowicz et al., 2017; Wang et al., 2005; Martín-Valdivia et al., 2008; Guyon et al., 2002; Breiman, 2001; Díaz-Uriarte & De Andres, 2006; Sneath & Sokal, 1973; Eisen et al., 1998).

Seven supervised machine learning FS methods and an unsupervised one were used in this study. Among the seven supervised machine learning FS methods, four filter approaches (Pearson Correlation Coefficient(PCC), Relief-F Feature Selection (Relief-F), Correlation Feature Selection (CFS) and Information Gain (IG)) were used in this study for their simplicity and high in good succession in performing FS. The features (genes) that selected by the filter approach are based on the relevance of the genes with the surrounding data (Chandrashekar & Sahin, 2014; Saeys, Inza & Larrañaga, 2007). In addition, one wrapper approach, Sequential Forward Selection (SFS), was carried out in this study as it acts as a searching algorithm that could find a subset of genes that is maximum according to the objective function (classification performance) (Chandrashekar & Sahin, 2014; Saeys, Inza & Larrañaga, 2007). Furthermore, this study also applied two embedded approaches, Support Vector Machine-based Recursive Feature Elimination (SVM-RFE) and Random Forest (RF), which embed variable selection as part of their learning process and evaluate them according to the classification result of specific machine learning method (Chandrashekar & Sahin, 2014; Saeys, Inza & Larrañaga, 2007). Hierarchical Clustering (HC) was applied as the unsupervised model in this study for its ability to select the genes without the knowledge of the class labels (Chandrashekar & Sahin, 2014).

As the supervised machine learning methods involves the classification according to the class labels, the normal samples were added back at stage (ii) so that the “learning procedures” of the machine learning can be completed.

The machine learning approaches applied in this study are more on exploratory models which the performances of the machine learning methods are not being emphasize here. The main objective of this stage is to obtain the optimum genes that commonly selected using different machine learning methods, regardless of the differences in the algorithm used.

The results (selected genes) of different FS and machine learning will be combined and converted in the scoring matrix. Scores will be determined and counted by the frequency of the genes that was being selected by various FS and machine learning methods.

Functional Pathway Analysis

Gene Set Enrichment Analysis (GSEA)

GSEA (http://software.broadinstitute.org/gsea/index.jsp) (Subramanian et al., 2005; Mootha et al., 2003) computationally evaluates the statistical significance of a priori defined set of genes with their biological states (i.e., phenotypes). All the genes in the datasets are ranked according to their differential expression between phenotype groups. In this study, normal tissue samples are also added into the dataset to enable the GSEA to calculate the Enrichment Scores (ES) between the phenotypes. A computational annotated gene set (C4) that consists of large collections of cancer-orientated microarray data is provided by Molecular Signature Database (MSigDB) and linked to GSEA in order to calculate the ES on our selected set of genes. ES is computed for every set of genes by going through the rank list from the top. A gene set will have a higher ES if the encountered gene in the gene input list is a member of the annotated gene set. A gene set is said to be significant if the normalized enrichment scores (NES) is high and its corresponding false discovery rate (FDR) is low. In this study, FDR value of <0.25 was set as the threshold to consider the significance of the gene enriched. Furthermore, pathway network could be visualized through Cytoscape (Shannon et al., 2003) software based on the GSEA results.

Result

Stage (i): gene expression analysis of individual dataset

LIMMA was used to perform the analysis in order to select out the top 10,000 differentially expressed genes from each of the individual dataset (Results shown in Appendix  S1).

LIMMA tests the average difference in log expression levels of two groups per gene by using moderated t-test. Therefore, genes selected by LIMMA are moderately different with average differential values, rather than extremely different.

Stage (ii): meta-analysis of multiple dataset

In the intersection process, 117 intersected genes among four cervical cancer datasets were identified. All cervical cancer samples from four datasets (total of 142 samples) were combined to form a meta-dataset with these 117 intersected genes. The meta-dataset is attached in Appendix S2.

Next, the analysis of the 117 genes by RankProd identified 32 upregulated genes (FC > 1, pfp < 0.01) and 33 down regulated genes (FC < 1, pfp < 0.01), with p-value < 0.05 and FDR of 0.05 as the cutoff points.

RankProd estimated the FDR for each gene and ranked the gene according to the value that has equal or smaller p-value in the whole list of genes (estimated percentage of false predictions). As the product of RankProd is affected more by extreme expression values, genes that are extremely expressed in at least one sample in the dataset tend to be selected. Tables 2 and 3 show the results of RankProd analysis.

Table 2 Results of RankProd analysis-32 upregulated genes.

ID_REF	Identifier	gene.index	RP/Rsum	FC: (class1/class2)	pfp	P.value	
206731_at	CNKSR2	102	2.697	1.13	7.77E−155	6.64E−157	
209243_s_at	PEG3	107	2.846	1.13	1.38E−149	2.36E−151	
210549_s_at	CCL23	110	4.156	1.19	3.15E−114	8.08E−116	
216108_at	LOC105373738	22	4.363	1.22	3.41E−110	1.17E−111	
220298_s_at	SPATA6	37	6.35	1.29	6.39E−81	2.73E−82	
212797_at	SORT1	20	6.371	1.33	9.09E−81	4.66E−82	
207996_s_at	LDLRAD4	71	10.32	1.66	1.72E−49	1.03E−50	
208762_at	SUMO1	59	11.02	1.66	8.37E−46	5.72E−47	
207257_at	EPO	84	12.25	1.48	4.26E−40	3.28E−41	
202035_s_at	SFRP1	73	12.53	1.50	5.99E−39	5.12E−40	
204672_s_at	ANKRD6	86	13.23	1.57	3.28E−36	3.09E−37	
220994_s_at	STXBP6	62	13.62	1.57	9.11E−35	9.34E−36	
213652_at	PCSK5	88	14.12	1.67	4.76E−33	5.29E−34	
213994_s_at	SPON1	50	14.35	1.61	2.72E−32	3.25E−33	
221606_s_at	HMGN5	35	15.39	1.68	5.13E−29	6.57E−30	
203440_at	CDH2	38	15.74	1,65	5.24E−28	7.17E−29	
207046_at	HIST2H4B	33	16.79	1.66	3.64E−25	5.29E−26	
211865_s_at	FZR1	99	19.24	1.69	1.47E−19	2.26E−20	
215401_at	AU147698	17	19.79	1.74	1.72E−18	2.79E−19	
213068_at	DPT	4	20.42	1.75	2.48E−17	4.24E−18	
214117_s_at	BTD	34	22.69	1.80	1.28E−13	2.30E−14	
34187_at	AK026407	100	23.65	1.83	2.78E−12	5.24E−13	
213611_at	AQP5	82	25.09	1.86	1.82E−10	3.58E−11	
209465_x_at	PTN	53	26.18	2.00	3.00E−09	6.16E−10	
208606_s_at	WNT4	13	27.42	1.92	5.31E−08	1.14E−08	
211814_s_at	CCNE2	52	33.28	2.00	0.001144	0.0002542	
208790_s_at	PTRF	81	33.49	2.00	0.00144	0.0003322	
200908_s_at	RPLP2	40	33.97	2.04	0.002479	0.0005933	
219795_at	SLC6A14	57	34.09	2.22	0.002761	0.0006842	
205730_s_at	ABLIM3	64	35.48	2.05	0.01187	0.003044	
202648_at	TCF3	32	37.13	2.10	0.05047	0.01337	
216267_s_at	TMEM115	95	37.72	2.07	0.07745	0.02118	

Table 3 Results of RankProd analysis-33 downregulated genes.

ID_REF	Identifier	gene.index	RP/Rsum	FC:(class1/class2)	pfp	P.value	
221475_s_at	RPL15	104	3.889	0.67	7.27E−120	1.24E−121	
203535_at	S100A9	103	4.437	0.71	1.20E−108	3.09E−110	
209719_x_at	SERPINB3	3	6.49	0.46	2.94E−79	1.00E−80	
200074_s_at	GUK1	55	7.095	0.45	4.17E−73	1.78E−74	
209351_at	KRT14	26	7.505	0.393	2.33E−69	1.20E−70	
201097_s_at	ARF4	11	8.908	0.32	2.43E−58	1.46E−59	
217845_x_at	HIGD1A	2	10.75	0.247	3.25E−47	2.22E−48	
210413_x_at	SERPINB4	7	10.93	0.27	2.59E−46	1.99E−47	
210835_s_at	CTBP2	77	12.68	0.17	2.39E−38	2.05E−39	
209720_s_at	SERPINB3	1	13.56	0.13	5.96E−35	5.60E−36	
200761_s_at	ARL6IP5	65	13.75	0.13	2.67E−34	2.74E−35	
201619_at	PRDX3	69	14.18	0.10	7.82E−33	8.69E−34	
206276_at	LY6D	47	14.37	0.15	3.27E−32	3.91E−33	
201098_at	COPB2	42	15.65	0.07	3.17E−28	4.07E−29	
201653_at	CNIH1	87	16.31	0.067	2.05E−26	2.80E−27	
211906_s_at	SERPINB4	8	16.42	0.003	3.80E−26	5.52E−27	
202753_at	PSMD6	56	17.6	0.01	3.54E−23	5.45E−24	
202209_at	LSM3	15	17.92	0.004	1.97E−22	3.20E−23	
211023_at	PDHB	9	17.95	0.018	2.14E−22	3.66E−23	
221896_s_at	HIGD1A	5	19.47	954.6	3.69E−19	6.62E−20	
213164_at	SLC5A3	63	22.18	787.9	2.11E−14	3.97E−15	
201863_at	FAM32A	106	22.74	767.2	1.42E−13	2.78E−14	
209694_at	PTS	66	24.92	688.4	1.09E−10	2.23E−11	
218845_at	DUSP22	10	26.22	675.4	3.21E−09	6.86E−10	
203315_at	NCK2	80	27.84	584.5	1.28E−07	2.84E−08	
213357_at	GTF2H5	114	28.09	612.8	2.09E−07	4.83E−08	
217850_at	SNORD19B	29	28.27	592.6	2.95E−07	7.06E−08	
218283_at	SS18L2	12	28.72	578.6	6.99E−07	1.73E−07	
203282_at	GBE1	14	31.25	526.7	5.77E−05	1.48E−05	
212488_at	COL5A1	68	32.86	426.3	0.0005592	0.000148	
219389_at	SUSD4	48	34.36	439.2	0.003389	0.000927	
218115_at	ASF1B	36	35.81	405	0.00481	0.004177	
201692_at	SIGMAR1	117	36.53	393	0.00775	0.008063	
Notes.

Description of Tables 2 and 3: gene.index-index of genes; RP/Rsum-computed Rank Product statistics; FC: (class1/class2)-average expression levels’ computed fold change under two conditions (upregulated class and downregulated class); pfp-estimated false positive predictions value of the genes; and P-value-estimated p-value of each gene.

Stage (iii): feature selection and machine learning methods

To find the optimum set of genes, eight FS and machine learning methods namely PCC, Relief-F, CFS, IG, SFS, SVM-RFE, RF and HC, were carried out using the set of genes resulting from stage (ii) of meta-analysis.

For HC analysis, the expression patterns of the 32 upregulated genes are shown in the heatmap and were further sub-clustered by cutting tree at the maximum heights of 1.5 of the row cluster. As a result, a cluster group that contained 20 genes was found to be most correlated to each other (Fig. 2A). The same rule was applied to the 33 downregulated genes. The most correlated group that contained 26 genes was also identified (Fig. 2B). The genes that were selected in the cluster that had the highest scores are presented in Tables 4 and 5 respectively.

As for the scoring measures, the final score obtained by each of the genes is the frequency it was selected by the listed FS and machine learning methods (1 score for selected, 0 score for not selected). The higher the score of a gene, the more frequently the gene was selected by the different FS methods, and the higher the potential for the gene to be a gene marker of cervical cancer. Tables 4 and 5 show the set of genes selected by the FS methods. The up- and downregulated genes with score ≥ 6 (full score = 8, threshold = 75%) were identified and selected. In total, 21 gene expression set consisted of 9 upregulated genes and 12 downregulated genes were selected.

Figure 2 Hierarchical clustering of the selected genes.

(A) Hierarchical clustering of the 32 upregulated genes. (B) Hierarchical clustering of the 33 downregulated genes. In the figures, each gene is represented by the rows and the samples are represented by the columns. The dendogram at the side indicates the relation between the pattern of the gene expression while the top-dendogram indicates the relation between the samples used. The level of expression of the genes, relative to the mean of the gene across all samples, is indicated by the color key, with the green representing the higher expression of the genes. The color bar represents the cluster of the genes after they have been cut at 1.5 of the height of the tree so that the clustering of the genes is seen more clearly. The color of the bar is indicated by the color key similarly.

Table 4 Ranking of each gene (upregulated) using proposed FS methods.

ID_REF	Identifier	HC	PCC	Relief-F	SFS	SVM-RFE	CFS	RF	IG	Score	
206731_at	CNKSR2	1	1	1	1	1	1	1	1	8	
209243_s_at	PEG3	1	0	1	1	1	1	1	1	7	
210549_s_at	CCL23	1	1	1	1	0	0	0	1	5	
216108_at	LOC105373738	1	1	1	0	0	1	0	1	5	
220298_s_at	SPATA6	1	1	1	1	1	0	1	1	7	
212797_at	SORT1	0	1	0	1	0	0	1	1	4	
207996_s_at	LDLRAD4	0	1	0	1	0	0	0	0	2	
208762_at	SUMO1	0	1	0	0	0	0	1	1	3	
207257_at	EPO	1	1	0	1	0	1	0	1	5	
202035_s_at	SFRP1	1	0	0	0	1	0	1	1	4	
204672_s_at	ANKRD6	0	0	1	1	1	0	1	1	5	
220994_s_at	STXBP6	0	0	1	1	1	0	1	1	5	
213652_at	PCSK5	1	0	0	0	1	1	0	0	3	
213994_s_at	SPON1	1	0	1	1	1	1	1	0	6	
221606_s_at	HMGN5	0	0	1	1	1	1	1	0	5	
203440_at	CDH2	1	0	1	1	0	1	0	1	5	
207046_at	HIST2H4B	1	0	0	0	0	0	0	0	1	
211865_s_at	FZR1	0	1	1	1	0	1	0	1	5	
215401_at	AU147698	1	0	1	0	1	1	0	1	5	
213068_at	DPT	1	0		0	0	1	1	1	5	
214117_s_at	BTD	1	1	0	1	1	1	1	1	7	
34187_at	AK026407	1	1	0	1	1	0	1	1	6	
213611_at	AQP5	1	0	0	1	1	1	1	0	5	
209465_x_at	PTN	0	0	1	0	1	1	0	0	3	
208606_s_at	WNT4	0	0	1	1	1	1	1	0	5	
211814_s_at	CCNE2	0	0	1	1	1	1	1	0	5	
208790_s_at	PTRF	0	0	0	0	0	1	0	0	1	
200908_s_at	RPLP2	1	1	1	1	1	0	1	1	7	
219795_at	SLC6A14	0	0	0	0	1	1	0	0	2	
205730_s_at	ABLIM3	1	0	1	0	1	1	1	0	5	
202648_at	TCF3	1	1	1	1	0	1	1	1	7	
216267_s_at	TMEM115	1	1	1	0	1	0	1	1	6	
Notes.

The nine upregulated genes selected by most (≥6) of the methods are CNKSR2, PEG3, SPATA6, SPON1, BTD, AK026407 RPLP2, TCF3 and TMEM115.

Table 5 Ranking of each gene (downregulated) using proposed FS methods.

ID_REF	Identifier	HC	PCC	Relief-F	SFS	SVM-RFE	CFS	RF	IG	Score	
221475_s_at	RPL15	1	1	1	0	1	1	1	1	7	
203535_at	S100A9	0	0	0	1	1	1	1	0	4	
209719_x_at	SERPINB3	0	0	0	0	1	1	0	0	2	
200074_s_at	GUK1	1	0	1	0	0	0	1	0	3	
209351_at	KRT14	0	0	1	0	0	1	0	0	2	
201097_s_at	ARF4	1	1	0	1	0	0	0	1	4	
217845_x_at	HIGD1A	1	1	1	1	1	0	1	1	7	
210413_x_at	SERPINB4	0	1	0	0	1	1	1	0	4	
210835_s_at	CTBP2	1	0	0	0	0	1	1	1	4	
209720_s_at	SERPINB3	0	1	0	1	0	0	1	0	3	
200761_s_at	ARL6IP5	1	1	1	1	1	0	1	0	6	
201619_at	PRDX3	1	1	1	1	1	1	0	1	7	
206276_at	LY6D	0	0	1	0	1	1	0	0	3	
201098_at	COPB2	1	1	1	1	1		1	1	7	
201653_at	CNIH1	1	1	1	1	0	1	1	1	7	
211906_s_at	SERPINB4	0	1	0	1	0	1	0	0	3	
202753_at	PSMD6	1	1	0	1	1	0	1	1	6	
202209_at	LSM3	1	1	1	1	1	1	0	0	6	
211023_at	PDHB	1	1	1	1	1	1	1	1	8	
221896_s_at	HIGD1A	1	1	1	1	1	0	1	0	6	
213164_at	SLC5A3	1	0	0	1	1	1	1	1	6	
201863_at	FAM32A	1	1	1	0	0	0	0		4	
209694_at	PTS	1	1	1	1	1	1	0	0	6	
218845_at	DUSP22	1	0	1	0	1	0	1	1	5	
203315_at	NCK2	1	0	0	1	1	0	1	1	5	
213357_at	GTF2H5	1	1	1	0	0	0	0	1	4	
217850_at	SNORD19B	1	1	0	1	0	1	0	1	5	
218283_at	SS18L2	1	1	0	0	0	0	0	1	3	
203282_at	GBE1	1	1	1	0	0	1	0	1	3	
212488_at	COL5A1	1	0	1	0	0		0	0	3	
219389_at	SUSD4	1	0	0	1	1	0	1	0	4	
218115_at	ASF1B	1	0	1	1	1	1	1	1	7	
201692_at	SIGMAR1	1	0	1	0	0	1	1	1	5	
Notes.

The 12 downregulated genes selected by most (≥6) of the methods are RPL15, HIGD1A, ARL6IP5, PRDX3, COPB2, CNIH1, PSMD6, LSM3, PDHB, SLC5A3, PTS and ASF1B. As gene HIGD1A is selected twice with different ID REF, it was only considered once here.

Functional Pathway analysis with GSEA

GSEA was carried out on the selected 21-gene expression set for functional analysis to measure the statistical significance of their biological phenotypes. In our study, we are assessing the enrichment of the selected 9 upregulated and 12 downregulated genes on their natural phenotypes, which are tumour and non-tumour. Therefore, samples of normal cells that were filtered at the beginning of the study were added back to complete the study. A computational gene set (C4) consisted of large collections of cancer-orientated microarray data was downloaded from MSigDB to calculate the enrichment scores of the nine upregulated genes and 12 downregulated genes among the annotated gene sets of C4. We evaluated the normalized enrichment scores (NES) and the corresponding false discovery rate (FDR) of each gene on all of the samples used in this study. In this study, a gene will be rejected when the FDR q-value is more than 0.25 (not significant). As NES values are dependent on the permutation method and size of the expression dataset, there is no standard threshold for this value in analysing the gene. A gene could match multiple sets of gene sets in the database, hence we selected the highest NES values for each respective gene as shown in Tables 6 and 7. The details of GSEA report on the annotation of the 21-gene expression set with the description of each gene sets are attached in Appendix S3.

Among the nine upregulated genes, six genes were matched with the cancerous gene set collections, as shown in Table 6. Notably, PEG3, SPON1, BTD and RPLP2 passed the defined threshold of FDR <0.25, indicating the potential of these genes in showing their significances in cervical cancer. As there was very limited information regarding genes AK026407, TCF3 and TMEM115, there are no matching genes with the MSigDB. Besides that, nine downregulated genes were matched with the C4 gene sets in MSigDB (Table 7). This showed that the nine genes were enriched and highly related to cancers. Among enriched genes, five downregulated genes, PRDX3, COPB2, LSM3, SLC5A3 and AS1B, passed the default threshold with FDR <0.25. These also provided some attraction points for the selected downregulated genes in proving their significances in cervical cancer. However, there are no matching of the genes ARL6IP5, CNIH1 and PTS with the MSigDB due to the low information found regarding these genes.

Figures 3 and 4 show the pathway network constructed according to the GSEA report generated in Appendix S3. The threshold that used in the pathway analysis was set to FDR <0.25 in order to standardize with GSEA. Each of the nodes in the networks represented the gene sets that the genes were involved in. The genes that showed enrichment with the gene sets were indicated in the colour nodes while the line connecting the nodes are the pathways that the enriched genes found in.

Table 6 Results of GSEA analysis using the nine upregulated genes.

Identifier	Highest normalized enrichment score (NES)	FDR q-value	Status	
CNKSR2	1.00	0.859	Rejected	
PEG3	1.33	0.000	Pass	
SPATA6	1.00	0.823	Rejected	
SPON1	1.60	0.053	Pass	
BTD	1.50	0.026	Pass	
AK026407	–	–	–	
RPLP2	1.40	0.000	Pass	
TCF3	–	–	–	
TMEM115	–	–	–	
Notes.

There was no matching of gene AK026407, TCF3 and TMEM115 with the MSigDB.

Table 7 Results of GSEA analysis using the 12 downregulated genes.

Identifier	Highest normalized enrichment score (ES)	FDR q-value	Status	
RPL15	1.21	0.779	Rejected	
HIGD1A	1.25	1.00	Rejected	
ARL6IP5	–	–	–	
PRDX3	1.41	0.020	Pass	
COPB2	1.41	0.020	Pass	
CNIH1	–	–	–	
PSMD6	1.44	0.560	Rejected	
LSM3	1.41	0.020	Pass	
PDHB	1.33	0.839	Rejected	
SLC5A3	1.50	0.000	Pass	
PTS	–	–	–	
ASF1B	1.43	0.000	Pass	
Notes.

There was no matching of gene ARL6IP5, CNIH1 and PTS with the MSigDB.

Figure 3 Pathway network of the upregulated genes.

Figure 4 Pathway network of the downregulated genes.

In Fig. 3, the upregulated genes PEG3, SPON1, BTD and RPLP2 were shown to be related with the pathway of gene expression, disease or cancer, and protein metabolism. Similarly, as shown in Fig. 4, downregulated genes PRDX3, COPB2, LSM3, SLC5A3 and ASF1B were involving in the gene expression, disease or cancer, protein metabolism and cell cycle pathways.

Discussion

Integrative approach of meta-analysis and machine learning

The aim of this study was to identify a set of more precise, accurate and reliable potential cervical cancer gene markers. Generally, there are three stages of analysis in this study: (i) gene expression analysis of individual dataset; (ii) meta-analysis of multiple datasets; and (iii) FS and machine learning analysis.

At the first stage, individual analysis of each cervical cancer microarray gene expression dataset was carried out using the LIMMA package in R. In LIMMA, the average difference of log expression levels between two groups of information provided for each gene was calculated using moderated t-statistic, to ensure that the final list of genes produced consists of genes that are consistently different between the groups. In other words, LIMMA avoids choosing the extremely different genes in the datasets. In this study, Benjamini and Hochberg’s method in controlling the FDR value was used as the default adjustment method of the adjusted p-value. The adjusted p-value is set to be <0.05 in expecting the proportion of the false discoveries in the selected gene group to be less than 5%.

In the meta-analysis stage, RankProd package was chosen for use for two reasons: to analyse the combination of datasets from different origins and to select genes with higher biological significances. RankProd package is able to avoid errors that are caused from batch and platform differences in combining expression values of various studies, by performing ranking of the genes by their differential expression (Hong et al., 2006). RankProd estimated the gene ranking by randomly permutating the list of genes. False discovery rates (FDR) for the whole list of genes which are equal or smaller than the p-value (known as false positive prediction, pfp) were calculated by this package and the genes were ranked accordingly.

The selection of genes in the RankProd analysis in stage (ii) was based on the criteria of FC > 1 (upregulated genes) or FC < 1 (downregulated genes). In log expression level, a gene is said to be significantly upregulated when the FC expression is >1 while it is significantly downregulated when the FC <1. An FC with value of two simply brings the meaning that the particular genes expressed twice more than the genes in the original control. In addition the php value is set to <0.01 in order to select the genes that sited on the top 1% of the truly significant genes among all the genes to be studied.

Following meta-analysis, 32 most significantly up-regulated and 33 down-regulated genes were selected.

To further narrow down the range of the genes selected and also to fully adopt the benefits from the statistical and also the machine-based analysis method, four filter approaches (PCC, Relief-F, CFS and IG), a wrapper approach (SFS), two embedded approach (SVM-RFE and RF) and an unsupervised approach of HC were carried out. As the algorithms in selecting genes by each method were different, the resultant genes selected by each method varies. In this study, most of the parameters set are based on the default values or the minimum values required as the focus is on the most significant genes that selected by the machine learning method and not the predictive performances of the machine learning methods).

We ensured the accuracy of our results by creating a scoring system to calculate the score of each gene. The threshold for the scoring system was set to 75%, where only the genes with score ≥6 (out of 8) would be selected. In total, 21-gene expression set which consisted of nine upregulated genes and 12 downregulated genes were selected. The upregulated genes that were most commonly selected by the stated FS methods were PEG3, SPON1, BTD, RPLP2, CNKSR2, SPATA6, AK026407, TCF3 and TMEM115 while the FS-commonly-selected-downregulated-genes were PRDX3, COPB2, LSM3, SLC5A3, ASF1B, RPL15, HIGD1A, ARL6IP5, CNIH1, PSMD6, PDHB, and PTS.

Functional pathway analysis- GSEA

According to the annotation of the 21 genes (nine upregulated genes and 12 downregulated genes) to the C4 cancerous gene set collection in MSigDB GSEA, it showed that six upregulated genes (CNKSR2, PEG3, SPATA6, SPON1, BTD and TCF3) and nine downregulated genes (RPL15, HIGD1A, PRDX3, COPB2, PSMD6, LSM3, PDHB, SLC5A3 and ASF1B) were enriched with the computational cancerous gene set. Particularly, four upregulated genes (SPON1, PEG3, RPLP2 and BTD) and five downregulate genes (PRDX3, COPB2, LSM3, SLCA3 and ASF1B) showed significant enrichment to the cancerous modules and cancerous genes neighbourhood gene sets with the FDR <0.25. This indicates that these nine genes were highly correlated with the formation of cancer and were selected as the nine potential gene expression signature in this study.

The pathway network (Figs. 3 and 4) were constructed using the GSEA report that attached as Appendix S3. The networks shown were based on the correlation between the gene sets. Similar with GSEA, the threshold was set to FDR <0.25 in this analysis. By comparing the map constructed and the GSEA result, it can be clearly seen that both results are supporting each other. The genes that showed significant enrichment with the C4 gene sets were shown and constructed in the network too. Moreover, the genes involved in the pathways of gene expression, disease or cancer, protein metabolism and cell cycle (Figs. 3 and 4). This further strengthened the role of the selected genes to be the nine potential signature in relation with cervical cancer development.

The nine potential signature will be discussed and supported by the previous studies done in the following section.

Upregulated genes are the genes that are known to be overexpressed during the gene regulation stage, causing the development of the tumour or potential cancerous cell. The most significant upregulated genes set that formed after the GSEA analysis are PEG3, SPON1, BTD and RPLP2. Table 8 shows the function of the upregulated genes selected, together with the related diseases and description.

Table 8 Function, related diseases and description of the most significant (upregulated).

Genes	Full name	Function	Related diseases and description	
PEG3	Paternally Expressed Gene 3	• protein coding gene.	• Hypermethylation of PEG3 is linked to epigenetics mechanism and HPV infection in cervical intraepithelial neoplasia (CIN) and Invasive Cervical Cancer. (Nye et al., 2013)
• Cell proliferation and p53-mediated apoptosis. (Yamaguchi et al., 2002)
• Involvement in the glioma and ovarian cancers (Jiang et al., 2010)	
SPON1	F-spondin 1	• extracellular matrix organization regulation, interaction between cells and axon guidance (Burstyn-Cohen et al., 1999)	• Showed extreme expression activities in the identification of colorectal biomarkers (Pagnotta et al., 2013)
• Down regulation activity in the cervical cell carcinoma (Zhai et al., 2010)	
BTD	bitonidase	• gene expression, proliferation and differentiation of cells, gene signaling (Faith & Abraham, 2013)	• gene expression signature that marks pelvic lymph node metastasis (PLNM) in cervical carcinoma (Huang et al., 2011)
• Involvement in thyroid and lung cancers (So et al., 2012; Scheerger & Zempleni, 2003)
• Potential serological marker in breast cancer plasma (Kang et al., 2010)	
RPLP2	Ribosomal Protein Lateral Stalk Subunit P2	• encodes a 60s subunit ribosomal protein.	• prognostic marker for gastric cancer (Zhang et al., 2011)
• involved in the carcinogenesis and progression of various cancers (Sharp et al., 1990)
• associated with other ribosomal protein genes with colorectal carcinomas (CRC) (Mao-De & Jing, 2007)	

Despite the focus of gene expression research in cancers are towards the upregulated genes, the role of downregulated genes should not be ignored. Lu et al. (2005) found that in general, tumours contained more downregulated miRNAs compared with the normal tissues, leading them to hypothesize that down-regulated expression of some miRNAs may play a key role in the development of tumours and that gene downregulation is as important an aspect to be studied as up-regulation.

The most significant identified downregulated genes in cervical cancer in this study are PRDX3, COPB2, LSM3, SLC5A3 and ASF1B. The biological functions, related diseases and description are shown in Table 9.

Table 9 Function, related diseases and description of the most significant genes (downregulated).

Genes	Full Name	Function	related diseases and description	
PRDX3	Peroxiredoxin3	• encodes antioxidant function protein, provides protection to mitochondria from oxidative stress (Safaeian et al., 2012)
• regulates the cellular Reactive Oxidative Species (ROS) level of the cell (Jiang, Sang & Qiu, 2017)	• showed correlation with the severity of the cervical carcinoma (Kim et al., 2009).
• Showed immunopositivity to be significantly higher in cervical cancer cells (Hu, Gao & Li, 2013)	
COPB2	Coatomer Protein Complex Subunit Beta 2	• protein coding gene.
• involves in the transduction of signal for G protein (Hu & Gao, 2012)	• One of the differentially expressed genes in LNCaP prostate cancer cell lines (Coutinho-Camillo et al., 2006)	
LSM3	Hypothetical protein LOC285378	• Involves in the process of MRNA splicing	• Involved in the rapid proliferation, invasiveness, oxidative phosphorylation and tumor size of cervical cancers (Lyng et al., 2006)	
SLC5A3	Solute Carrier Family 5 Member 3	• Involves in the osmoregulation of cells (Wright & Turk, 2004)
• prevents the impairment of cell cellular function (Database GCHG, 2017)	• differentially expressed between Parental SiHa Cells and SiHa/R Cells of cervical cancer (Chung et al., 2005)
• involved in the formation of fusion transcripts gene in some prostate cancer cases (Narod, Seth & Nam, 2008)	
ASF1B	Anti-Silencing Function 1B Histone Chaperone	• codes for the substrate protein of the cycle regulated-kinase cell.
• derepress the overexpression of transcriptional silencing (Le et al., 1997)	• associated with the aggressiveness of breast tumor (Corpet et al., 2011)
• leaded to the poor disease outcome in the study of the cervical cancer proliferation cluster that corresponded to the 163 transcripts (Rosty et al., 2005).	

Our integrated approach of meta-analysis and machine-learning method successfully identified the potential genetic markers in cervical cancer, along with the evidence provided from the GSEA results and the previous study done. The identified genes can be considered as potential gene signatures in cervical cancer. However, further evaluation using other experimental methods are needed in order to validate the significance of the nine potential-gene expression cervical cancer signatures.

Conclusion

In this study, we proposed an integrative machine learning analysis of multiple gene expression profiles in cervical cancer. This integrative analysis is divided into three stages: (i) gene expression analysis of individual dataset; (ii) meta-analysis of multiple datasets; and (iii) FS & machine learning analysis with PCC, Relief-F, CFS, IG, SFS, SVM-RFE, RF and HC. As a result, nine upregulated genes and 12 downregulated genes, were selected through the three-step analysis. The result of the functional pathway analysis in GSEA showed the significant enrichment in nine potential gene expression signature namely PEG3, SPON1, BTD, RPLP2 (as upregulated genes) and PRDX3, COPB2, LSM3, SLC5A3, ASF1B (as downregulated genes). More experiments are needed in order to confirm the significance of the selected nine potential gene expression signature in cervical cancer.

Supplemental Information

Appendix S1 Result of LIMMA analysis

Click here for additional data file.

Appendix S2 Gene expression dataset of combined dataset

Click here for additional data file.

Appendix S3 GSEA analysis result

Click here for additional data file.

Additional Information and Declarations

Competing Interests

Author Contributions

Patent Disclosures

The authors declare there are no competing interests.

Mei Sze Tan conceived and designed the experiments, performed the experiments, analyzed the data, prepared figures and/or tables, authored or reviewed drafts of the paper, approved the final draft.

Siow-Wee Chang conceived and designed the experiments, performed the experiments, analyzed the data, contributed reagents/materials/analysis tools, prepared figures and/or tables, authored or reviewed drafts of the paper, approved the final draft.

Phaik Leng Cheah conceived and designed the experiments, analyzed the data, contributed reagents/materials/analysis tools, prepared figures and/or tables, authored or reviewed drafts of the paper, approved the final draft.

Hwa Jen Yap conceived and designed the experiments, contributed reagents/materials/analysis tools, authored or reviewed drafts of the paper, approved the final draft.

The following patent dependencies were disclosed by the authors:

All the data used are available from the National Center Biotechnology Information Gene Expression Omnibus (NCBI GEO) (http://www.ncbi.nlm.nih.gov/geo) with accession numbers GSE26511, GSE5787, GSE9750, GSE63514.

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
