# Peer review of "Integrative machine learning analysis of multiple gene expression profiles in cervical cancer"

_PeerJ, doi:10.7717/peerj.5285_

## Round 0.1 · original submission · Major Revisions

This paper analyzes gene expression datasets from cervical cancer microarray data. The main issue with the current manuscript is that there are too many different analytical tools employed and it is not clear why and how exactly the analyses are done. Figure 1 presents a nice overview, but the main text does not complement information in Figure 1. Instead of describing what is the definition of each standard method, please provide information about what is your objective in each step, why you chose certain method, how you implemented it, and what are the results from the analysis. Current manuscript is just a collection of description for each component of Figure 1 and it lacks the overall story and synthesis.

·

Basic reporting

"Machine Learning" is a very broad term. I suggest the authors to be more specific than the current form on the Abstract section. Whether it is supervised or unsupervised learning? Is it predictive or exploratory?
Figure 1 is nicely drawn for a bird-eye view on the manuscript.

Experimental design

Sufficient experimental design for "PeerJ".

Validity of the findings

The last sentence in the abstract looks troublesome: "... Nevertheless, further validations are needed in order to validate the significance of selected 9-potential-gene expression signature in cervical cancer." It significantly weakens the scientific soundness. In addition, such a sentence is also repeated in the conclusion.

Additional comments

In general, the authors have written a reasonable manuscript for PeerJ. It has the basic components in computational genome studies. I have the following suggestions:

Since the study is a data integration study. Special care has to be taken for data normalization among those input datasets. The authors should carefully write down most of the parameter setting for readers. Otherwise, readers may not be convinced.

From line 118 to 200, the authors have described the off-the-book-shelf concepts in bioinformatics. I doubt if it is necessary for the authors to repeat those basic knowledge on this scientific research manuscript?

It would be interesting if the authors can describe their works in the context of pathway network analysis on top of GSEA.

Reviewer 2 ·

Basic reporting

no comment

Experimental design

no comment

Validity of the findings

How to evaluate the biomarkers identified by the model?

Additional comments

In this manuscript, the authors applied the existing machine learning methods on four Cervical cancer microarray gene expression datasets to identify the differentially expressed genes. Then the GSEA enrichment analysis was applied on the genes to identify the potential biomarker for Cervical cancer. Overall, the analyses are standard. The contribution to the computation is limited.

Major comments:
(1) Detailed description of integrating four datasets is required. It is very hard to follow how to do the intersection process. All the top 10000 genes are differentially expressed genes?
(2) Rank Product did the differential expression analysis to select the up-regulated and down-regulated genes. It is not necessary to do the feature selection again by using the other methods.
(3) Most of the eight machine learning methods are not comparable, e.g., HC is an unsupervised machine learning model to clustering the features and SVM and RF are supervised machine learning models to classify the samples. In additional, how to select features by applying RF?

---

## Round 0.2 · Major Revisions

A reviewer has raised concerns about the manuscript, mainly regarding the flow of the manuscript. I think the manuscript needs better description about the 'overall picture' of the proposed framework, rather than explaining about individual standard analysis tools. The reviewer also asked question about gene set enrichment analysis, which I believe is written slightly confusing way. 40,000 genes would not mean 40,000 'different' genes, I believe, but it's not clear in the context. Also, it's not clear whether you counted isoforms and transcripts separately or not. Another concern for GSEA is that FDR<0.25 would be too loose for a significant threshold. Please improve the structure of the manuscript and revise GSEA part.

·

Basic reporting

The authors have addressed my concerns.

Experimental design

The authors have addressed my concerns.

Validity of the findings

The authors have addressed my concerns.

Additional comments

The authors have addressed my concerns.

Reviewer 2 ·

Basic reporting

no comment

Experimental design

no comment

Validity of the findings

no comment

Additional comments

Thank you for addressing my questions in the rebuttal letter. I have the following three questions:
(1). There are only 21,000 ~ 22,000 human genes, how to get around 40,000 genes in your analysis? Biologically, it is not convincing that the top 10,000 genes are differentially expressed.
(2) Again,I still feel it is misleading to list eight incomparable methods in the manuscript. If the eight algorithms are not comparable, how could the scores calculated for each gene in eight algorithms are comparable? In addition, what is the computation contribution of this manuscript if all the eight algorithms packages are available in R or Python?
(3) Whether the findings are reducible on the RNA-Seq data?

---

## Round 0.3 · accepted · Accept

Please note that PeerJ does not provide any editorial services, so it is up to the authors to make language corrections before the paper is published online.

Reviewer 2 ·

Basic reporting

no comment

Experimental design

no comment

Validity of the findings

no comment

Additional comments

The authors have addressed all of my comments and I recommend to accept the paper.